# Outcomes of Transcatheter Edge-to-Edge Repair in Degenerative vs. Functional Mitral Regurgitation

**DOI:** 10.3390/jcm11206010

**Published:** 2022-10-12

**Authors:** Mark Kheifets, Filippo Angelini, Fabrizio D’Ascenzo, Stefano Pidello, Haya Engelstein, Pier Paolo Bocchino, Paolo Boretto, Simone Frea, Amos Levi, Hana Vaknin-Assa, Mordehay Vaturi, Yaron Shapira, Ran Kornowski, Leor Perl

**Affiliations:** 1Division of Cardiology, Rabin Medical Center, 39 Ze’ev Jabotinsky St., Petach Tikva 4941492, Israel; 2Faculty of Medicine, Tel Aviv University, Tel Aviv 69978, Israel; 3Division of Cardiology, Cardiovascular and Thoracic Department, Città della Salute e della Scienza Hospital, 10126 Turin, Italy; 4Division of Cardiology, Department of Medical Sciences, University of Turin, 10124 Turin, Italy; 5The Adelson School of Medicine, Ariel University, Ariel 4076414, Israel

**Keywords:** functional mitral regurgitation (FMR), degenerative mitral regurgitation (DMR), transcatheter edge-to-edge repair (TEER)

## Abstract

Current guidelines support the use of transcatheter edge-to-edge repair (TEER) for patients with both primary and secondary mitral regurgitation. We aimed to compare the prognoses of TEER in degenerative mitral regurgitation (DMR) vs. functional mitral regurgitation (FMR). A total of 208 consecutive patients who underwent TEER over a ten-year period were analyzed. Primary endpoints included rates of all-cause death and major adverse cardiac events (MACE: composite of all-cause death, hospitalizations for heart failure, mitral valve surgery, or TEER re-intervention). A total of 148 (71%) patients were identified with FMR, while 60 (29%) were identified with DMR. Patients in the FMR group were younger (77.2 ± 8.4 vs. 80.2 ± 7.2, *p* = 0.02), suffered more frequently from coronary artery disease (54.1% vs. 10.0%, *p* = 0.02), and atrial fibrillation/flutter (70.9% vs. 38.3%, *p* = 0.02). Rates of 1-year death (21.6% vs. 10.0%, *p* = 0.03) and MACE (41.2% vs. 21.7%, *p* = 0.02) were higher for the FMR group, as compared to the DMR group. After correcting for variables, FMR independently predicted rates of MACE (HR-1.78, 95% CI 1.23–2.48, *p* = 0.04) and had a non-significant effect on one-year mortality (HR-1.67, 95%CI 0.98–3.74, *p* = 0.07). In our experience, worse overall 1-year composite MACE outcomes were observed after TEER in patients with FMR as compared to patients with DMR.

## 1. Introduction

Mitral regurgitation (MR) is the most common heart valve disease in the global population [1]. Even when isolated or untreated, MR is associated with notable heart failure and/or sizeable excess mortality. This holds true in all possible subsets and classifications of MR. Despite these poor outcomes, only a minority of affected patients undergo mitral valve surgery [2]. Degenerative mitral regurgitation (DMR) originates from a structural or degenerative abnormality of the mitral valve apparatus, while functional mitral regurgitation (FMR) is usually caused by a disease of the left ventricle, which results in malcoaptation of the mitral leaflets [3]. Less frequently, FMR develops as a result of atrial fibrillation (AF) and/or heart failure with preserved ejection fraction (HFpEF), which is defined as atrial FMR. This unique entity is both under-recognized and under-reported [4]. Surgical treatment is still the gold standard for both symptomatic DMR and FMR, but mitral transcatheter edge-to-edge repair (TEER) is a safe and well-established treatment option for patients with both DMR and FMR [5,6,7]. The indications for mitral valve intervention are less clear for FMR than for primary DMR [3]. In contrast to the COAPT trial [7], the MITRA-FR trial [8] failed to demonstrate significant differences in short- or long-term outcomes between TEER and guidelines-directed medical therapy (GDMT), compared to GDMT alone, in patients with FMR. Thus far, meta-analyses have failed to demonstrate significant differences in short- or long-term outcomes between patients with DMR or FMR who underwent TEER procedures, despite FMR patients having a higher patient-centered risk profile [9,10]. In the current study, we therefore aimed to compare both short- and long-term prognoses of the TEER procedure in DMR vs. FMR in a cohort of MR patients and identify the predictors for successful and sustainable interventions in these two different MR conditions.

## 2. Methods

### 2.1. Patients and Data Collection

In this prospective registry, included were all patients undergoing TEER at the Rabin Medical Center, Petah-Tikva, Israel and the Città della Salute e della Scienza Hospital, Turin, Italy for the first time using the MitraClip percutaneous mitral valve repair (Abbott Vascular, Inc., Santa Clara, CA, USA) between January 2012 and May 2021. All patients suffered from severe symptomatic MR. The data is continuously entered into an ongoing prospective registry for purposes of recording and monitoring patient-related parameters, clinical events, and angiographic findings. Exclusion criteria were immediate conversion to surgery due to technical failure, unavailability of echocardiographic data after discharge, concomitant transcatheter tricuspid repair, active malignancy, systemic infection, or cardiogenic shock at baseline. The study was conducted in accordance with the ethical principles that have their origin in the Declaration of Helsinki and was approved by the ethics committees of both medical centers.

### 2.2. Endpoints

As recently validated and adopted in several medical centers worldwide [11,12,13], a successful TEER procedure was defined as mild or less residual MR (1+). The co-primary endpoints were rates of all-cause death and major adverse cardiac events (MACE, which comprised: all-cause death, hospitalizations for heart failure, mitral valve surgery or TEER re-interventions) at a 12-month follow-up. Patients were treated by mitral valve surgery or TEER re-intervention during follow-up if recurrent symptomatic moderate MR or above, amenable to the appropriate therapy, was demonstrated. Secondary outcomes included the individual components of MACE and residual MR, defined as grade 2 MR or above. Additionally, we assessed the independent determinants of the occurrence of death and MACE at 1 year using multivariable analysis.

### 2.3. Statistical Analysis

The Kolmogorov–Smirnov and Shapiro–Wilk tests were used to investigate the normality of the distribution of continuous variables. Continuous variables following a normal distribution are reported as mean ± standard deviation and were compared using the Student’s *t*-test (paired or unpaired), whereas those not following a normal distribution are presented as median and interquartile range and were compared using the Mann–Whitney U test. Categorical variables are reported as counts and percentages and were compared using the chi-squared or Fisher’s exact test, as appropriate.

The survival rate free from clinical endpoints was estimated using the Kaplan–Meier method, and the differences between groups were calculated using the log-rank test. A Cox regression analysis was performed to identify the multivariable predictors of mortality and MACE at 12 months. Variables with *p* < 0.25 on univariate analysis were included in the final multivariable model. Each result is expressed as a hazard ratio (HR) and a corresponding 95% confidence interval (CI). A 2-sided *p* value of < 0.05 was considered statistically significant for all analyses. All statistical analyses were performed using SPSS version 28.0 (IBM Corp., Armonk, NY, USA).

## 3. Results

### 3.1. Patient and Echocardiographic Characteristics

A total of 226 patients had undergone TEER over the study period (2012–2021), of whom 18 were excluded due to conversion to surgery and missing echocardiographic data for the complete follow-up period (Figure 1).

Of the remaining eligible 208 patients, 148 (71%) were identified with FMR, while 60 (29%) were identified with DMR. Baseline characteristics of DMR vs. FMR are shown in Table 1. Patients in the FMR group were younger (77.2 ± 8.4 vs. 80.2 ± 7.2, *p* = 0.02), suffered more frequently from coronary artery disease (CAD) (54.1% vs. 10.0%, *p* = 0.02), and atrial fibrillation/flutter (70.9% vs. 38.3%, *p* = 0.02), as compared to the DMR group.

Baseline echocardiographic characteristics are shown in Table 2. Patients in the FMR group had higher left ventricular end diastolic diameter (LVEDd) (53.8 ± 7.8 vs. 50.9 ± 6.9 mm, *p* = 0.04), higher left ventricular end systolic diameter (LVESd) (38.2 ± 9.8 vs. 34.5 ± 8.4 mm, *p* = 0.03), lower left ventricular ejection fraction (LVEF) (42.4 ± 12.1 vs. 52.1 ± 10.9, *p* < 0.01), lower MR volume (60.4 ± 22.3 vs. 68.1 ± 23.1 mL, *p* = 0.04), and higher pulmonary artery pressure (53.2 ± 14.4 vs. 50.5 ± 14.1 mmHg, *p* = 0.04), as compared to the DMR group.

### 3.2. Clinical Outcomes

Clinical outcomes during study follow-up are shown in Table 3. There were no statistically significant differences in MR grade during the immediate post-procedural period between the two groups, nor were there any significant differences in rates of residual MR at both 1 month and 1 year between the two groups. Furthermore, there were no statistically significant differences in rates of death, MACE, hospitalizations for heart failure, or re-intervention at 1 month between both groups. However, rates of death (21.6% vs. 10.0%, *p* = 0.04), MACE (41.2% vs. 21.7%, *p* = 0.02), and hospitalizations for heart failure, (30.4% vs. 15.0%, *p* = 0.03) were all significantly higher in the FMR group after 1 year as compared to the DMR group.

Kaplan–Meier analysis for death and MACE are shown in Figure 2 and Figure 3 (respectively).

The results of the Cox regression analysis for death are shown in Table 4. After adjusting for possible variables, age (HR-1.07, 95% CI 1.02–1.92, *p* = 0.04) emerged as an independent risk factor for mortality at 1 year, while higher LVEF (HR-0.93, 95% CI 0.87–0.97, *p* = 0.02) was found to be an independent protective factor for mortality at 1 year. Although not statistically significant, FMR etiology (HR-1.67, 95% CI 0.98–3.74, *p* = 0.07) displayed a numerical trend towards higher 1-year mortality.

For MACE (Table 5), after adjustment for possible confounders, age (HR-1.10, 95% CI 1.01–1.24, *p* = 0.02), FMR etiology (HR-1.78, 95% CI 1.23–2.48, *p* = 0.04), and post-procedure MR (HR-1.96, 95% CI 1.09–2.77, *p* = 0.03) emerged as independent risk factors for 1 year events, while higher LVEF (HR-0.92, 95% CI 0.84–0.98, *p* = 0.04) was found to be an independent protective factor.

The Kaplan–Meier curves for the adjusted death and MACE are shown in Figure 4.

## 4. Discussion

According to our decade-long clinical experience with TEER for both FMR and DMR, we observed higher rates of both death and MACE at 1 year in the FMR group, as compared to the DMR group. Following adjustment, FMR was independently associated with excess MACE but had a marginal, non-significant effect on 1-year mortality following TEER—probably due to the relatively small number of mortality events. Multivariable analysis identified age as an independent risk factor for both mortality and MACE at 1 year, whereas higher LVEF was found to be an independent protective factor for both mortality and MACE at 1 year. In addition, FMR etiology and post-procedural residual MR emerged as independent risk factors for MACE at 1 year.

Our findings are consistent with earlier research. As was shown by Sorajja et al. [14], patients who underwent TEER for DMR had lower rates of both all-cause mortality and re-hospitalizations for heart failure as compared to patients with FMR who underwent TEER. As was later described by Kar et al. [15], patients with DMR who underwent TEER procedures were older, had lower incidence of CAD, better LVEF, and smaller ventricles as compared to patients with FMR who underwent TEER. Moreover, the recently proposed MitraScore by Raposeiras-Roubin et al. [16] identified reduced LV function as an independent risk factor for all-cause mortality following TEER for MR. Finally, as was described by Yoon et al. [17], patients who underwent TEER for FMR had higher rates of atrial fibrillation, lower LV ejection fraction, and larger LV dimensions, and demonstrated higher rates of both all-cause mortality and re-hospitalizations for heart failure, as compared to patients with DMR.

Several reasons may explain these findings: First, there are fundamental anatomical and pathophysiological differences between the valve characteristics of patients suffering from DMR and FMR. DMR may be caused by perforation or cleft (Carpentier type I) or leaflet/chordal thickening diseases (Carpentier type IIIA). Compared to patients with FMR, patients with DMR have a larger regurgitant volume, moderate cardiac remodeling, a normal stroke volume index, and mildly elevated pulmonary pressure. On the other hand, the two main etiologies of FMR are annular dilation or atriogenic MR (Carpentier type I) and leaflet tethering from a ventricular disease (Carpentier IIIB). Second, FMR due to LV dilation and systolic dysfunction, often with concomitant mitral annular dilation, is a common consequence of either ischemic or nonischemic cardiomyopathies. A number of mechanisms may contribute to malcoaptation of the mitral valve leaflets in FMR: (1) global and/or regional LV dilation/dysfunction that decreases the closing forces of the leaflets; (2) displacement of the papillary muscles with tethering of the leaflets into the ventricular cavity, which outweighs the closing forces; (3) dilation and dysfunction of the annulus; and (4) inadequate leaflet adaptation to ventricular or atrial enlargement. In the setting of DMR, the left ventricular end diastolic volume (LVEDV) increases, which results in increased left ventricular (LV) wall stress, a stimulus for eccentric hypertrophy. Initially, the reduced afterload is associated with an increased LV ejection fraction (LVEF). However, prolonged hemodynamic overload ultimately leads to myocardial decompensation. Reduction of MR abolishes the hemodynamic burden responsible for the deterioration in LV function and may restore it if it is already depressed. The anatomic features associated with these mechanisms thus predict the severity of MR and recurrence after surgical or transcatheter repair [3].

While initial landmark trials focused mostly on patients with DMR [18], real-world experience shows FMR has become the principal indication for TEER in most medical centers worldwide [19,20,21]. Importantly, the pivotal COAPT trial [6] and its subsequent three-year outcomes follow-up study [22] demonstrated improved survival, quality of life, and functional capacity among patients on GDMT who underwent TEER, as compared to patients treated with GDMT alone for 36 months. In contrast, both the MITRA-FR [8] and its subsequent two-year outcomes follow-up study [23], failed to demonstrate any significant reduction in either death or hospitalization for heart failure among patients on GDMT who underwent TEER, as compared to patients treated with GDMT alone. Importantly, a recent 12 month landmark analysis showed a non-statistically significant numerical reduction in the cumulative rate of HF hospitalizations between 12 and 24 months [24]. Although conflicting at first glance, the outcomes of these studies might be explained by the seemingly different patient populations [25]. However, little is known regarding the outcomes of patients with FMR treated by TEER as compared to patients with DMR. Therefore, our study sheds important light on real-world experience with TEER.

The field of transcatheter mitral valve repair is rapidly evolving. In addition to the well-established TEER techniques, there is also growing evidence for therapeutic strategies targeting the different parts of the mitral apparatus (mitral valve annulus, mitral valve leaflets, and mitral valve chordae) [26]. As more data is collected from this ongoing clinical experience, we expect to learn more regarding the outcomes of patients with MR treated by transcatheter methods, as well as the appropriate methods to treat patients with both FMR and DMR.

## 5. Limitations

This study has several limitations. First, although all data were collected prospectively, the observational design of the study has inherent limitations associated with a non-randomized comparison. Second, out of the 208 patients, only 60 (29%) suffered from DMR. Lastly, we have begun collecting data before the 5-point scale, which includes massive and torrential sub-categories within the previous definition of severe TR [27]. We have based our TR estimations according to both the older European Association of Cardiovascular Imaging (EACVI) and American Society of Echocardiography (ASE) recommendations, where “Severe TR” is defined quantitatively as an effective regurgitant orifice area (EROA) of ≥40 mm^2^ and a regurgitant volume of ≥45 mL [28,29].

## 6. Conclusions

Our study suggests worse overall outcomes for FMR, as compared to DMR, following the TEER procedure. Higher rates of both death and MACE were observed in the FMR group, mainly driven by heart failure hospitalizations. These findings further strengthen the importance of optimal GDMT after interventional procedures. Large, randomized studies, with extended follow-up, are needed to fully understand the effects of TEER on these two very different MR etiologies.

## Figures and Tables

**Figure 1 jcm-11-06010-f001:**
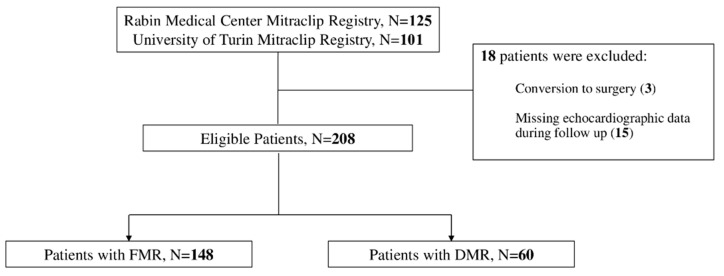
Patient Flow Chart.

**Figure 2 jcm-11-06010-f002:**
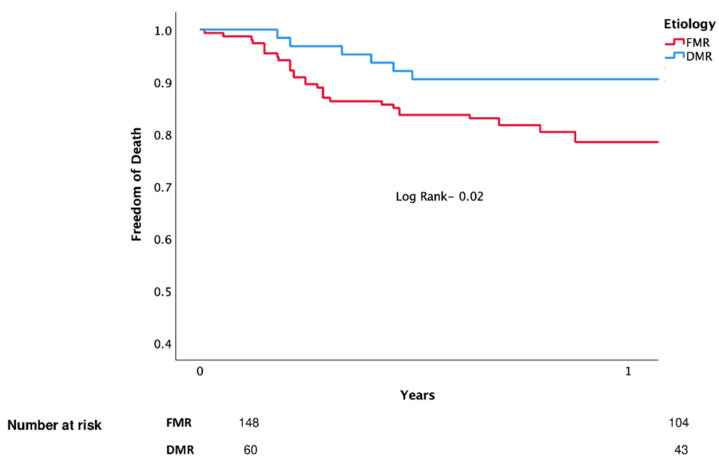
Death (unadjusted).

**Figure 3 jcm-11-06010-f003:**
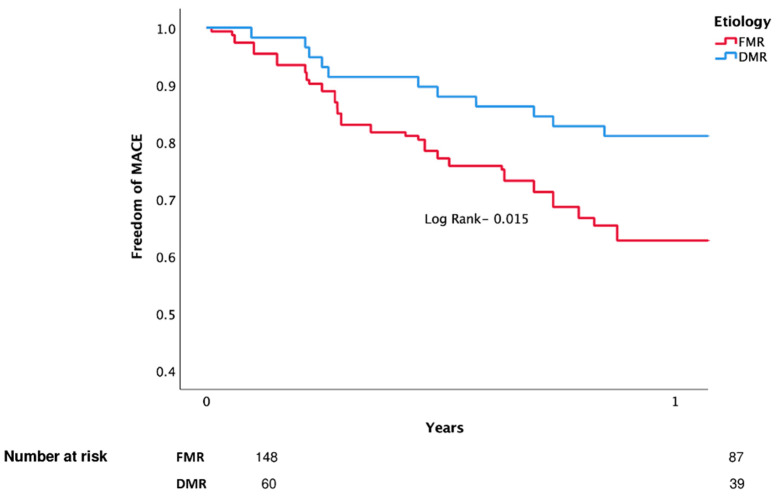
MACE (unadjusted).

**Figure 4 jcm-11-06010-f004:**
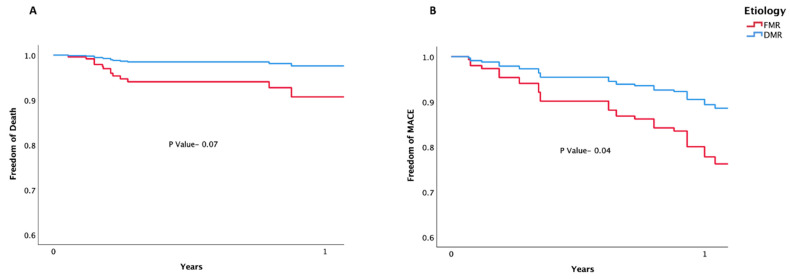
(**A**) Death (adjusted) and (**B**) Mace (adjusted).

**Table 1 jcm-11-06010-t001:** Baseline Characteristics.

Variable	FMR N = 148 (%)	DMR N = 60 (%)	*p* Value
Age	77.2 ± 8.4	80.2 ± 7.2	0.02
Gender-Female	47 (31.8)	19 (31.7)	0.26
BMI	25.4 ± 3.1	26.1 ± 4.2	0.21
Hemoglobin (g/dL)	11.6 ± 1.7	12.2 ± 1.2	0.11
GFR (mL/min)	58.4 ± 27.2	58.1 ± 20.1	0.36
Hypertension	118 (79.8)	38 (63.3)	0.42
Diabetes Mellitus	71 (48.0)	21 (35.0)	0.27
Dyslipidemia	101 (68.2)	43 (71.7)	0.10
Smoking	38 (25.7)	26 (43.3)	0.31
COPD	24 (16.2)	11 (18.3)	0.23
Stroke (CVA or TIA)	36 (24.3)	8 (13.3)	0.09
Coronary Artery disease	80 (54.1)	6 (10.0)	0.02
Cardiac Surgery	47 (31.8)	6 (10.0)	0.09
Atrial Fibrillation/Flutter	105 (70.9)	23 (38.3)	0.02
Pacemaker	31 (20.9)	13 (21.7)	0.41
Oncologic disease	17 (11.5)	12 (20.0)	0.12
Previous MitraClip	4 (2.7)	1 (1.7)	0.22
Peripheral pitting edema	21 (14.2)	9 (15.0)	0.34
NYHA class - no. (%)			0.18
II	42 (28.4)	21 (35.0)	
III	60 (40.5)	23 (38.3)	
IV	46 (31.1)	16 (26.7)	
STS score- for MV repair mortality	5.8 ± 4.1	5.4 ± 3.3	0.37
Medications			
Antiplatelets	67 (45.3)	20 (33.3)	0.12
Anticoagulants	38 (25.7)	19 (31.7)	0.29
ACEi/ARB	100 (67.6)	41 (68.3)	0.32
Beta blockers	126 (85.1)	47 (78.3)	0.39
MRA	42 (28.4)	12 (20.0)	0.42

Values are mean ± SD or n (%). Abbreviations: FMR, functional mitral regurgitation; DMR, degenerative mitral regurgitation; BMI, body mass index; GFR, glomerular filtration rate; COPD, chronic obstructive pulmonary disease; CVA, cerebrovascular accident; TIA, transient ischemic attack; NYHA, New York heart association; STS, society of thoracic surgery; ACEi, angiotensin-converting enzyme inhibitor; ARB, angiotensin receptor blocker; MRA, mineralocorticoid receptor antagonist.

**Table 2 jcm-11-06010-t002:** Baseline Echocardiographic Characteristics.

Variable	FMR N = 148 (%)	DMR N = 60 (%)	*p* Value
LVEDd (mm)	53.8 ± 7.8	50.9 ± 5.1	0.03
LVESd (mm)	38.2 ± 9.8	34.5 ± 6.9	0.04
IVS (mm)	10.8 ± 4.2	10.7 ± 3.9	0.62
PWT (mm)	10.2 ± 5.4	10.0 ± 5.2	0.42
Simpson EF (%)	42.4 ± 12.1	52.1 ± 10.9	<0.01
Mitral anulus calcification	27 (18.2)	8 (13.3)	0.42
Effective regurgitant orifice area (cm^2^)	0.4 ± 0.1	0.4 ± 0.2	0.82
MR Volume (mL)	60.4 ± 22.3	68.1 ± 23.1	0.04
Tricuspid Regurgitation Severity			
No TR (0)	23 (15.5)	5 (8.3)	0.24
Mild TR (1)	107 (72.3)	42 (70.0)	
Moderate TR (2)	10 (6.8)	11 (18.3)	
Severe TR (3)	8 (5.4)	2 (3.3)	
Estimated PA pressure (mmHg)	53.2 ± 14.4	50.5 ± 14.1	0.04
Right ventricular dysfunction (Grade)			0.30
Normal (0)	82 (55.4)	42 (70.0)	
Mild (1)	61 (41.2)	16 (26.7)	
Moderate (2)	5 (3.4)	2 (3.3)	
Left Atrial Area (cm^2^)	30.7 ± 8.1	31.6 ± 9.0	0.27

Values are mean ± SD or n (%). Abbreviations: FMR, functional mitral regurgitation; DMR, degenerative mitral regurgitation; LVEDd, left ventricular end diastolic diameter; LVESd, left ventricular end systolic diameter; IVS, intraventricular septum, PWT, poster wall thickness; EF, ejection fraction; MR, mitral regurgitation; TR, tricuspid regurgitation; PA, pulmonary artery.

**Table 3 jcm-11-06010-t003:** Clinical outcomes during follow-up.

Variable	FMR N = 148 (%)	DMR N = 60 (%)	*p* Value
Immediate post-procedure			
MR up to grade 1	139 (93.9)	55 (91.7)	0.83
MR grade 2	9 (6.1)	5 (8.3)	
MR grade 3	0 (0.0)	0 (0.0)	
MR grade 4	0 (0.0)	0 (0.0)	
1-month			
MR grade 2 or above	7 (4.7)	4 (6.7)	0.47
Hospitalization for HF	10 (6.8)	2 (3.3)	0.15
Surgery or clip re-intervention	5 (3.4)	2 (3.3)	0.89
MACE	15 (10.1)	3 (5.0)	0.09
Death	8 (5.4)	1 (1.7)	0.12
1-year			
MR grade 2 or above	13 (8.8)	5 (8.3)	0.17
Hospitalization for HF	45 (30.4)	9 (15.0)	0.03
Surgery or clip re-intervention	7 (4.7)	3 (5.0)	0.08
MACE	61 (41.2)	13 (21.7)	0.02
Death	32 (21.6)	6 (10.0)	0.04

The values are mean ± SD or n (%). Abbreviations: FMR, functional mitral regurgitation; DMR, degenerative mitral regurgitation; MR, mitral regurgitation; HF, heart failure; CVA; cerebrovascular accident; MACE, major adverse cardiovascular events.

**Table 4 jcm-11-06010-t004:** Cox regression for Death.

Variable	HR (95% CI)	*p* Value
Age (years)	1.07 (1.02–1.92)	0.04
STS score	1.03 (0.89–1.27)	0.79
Diabetes Mellitus	1.39 (0.72–4.21)	0.23
GFR (mL/min)	0.97 (0.94–1.01)	0.07
LVEF (per 1%)	0.93 (0.87–0.97)	0.02
Functional Class (NYHA)	1.08 (0.84–2.93)	0.42
Right ventricular dysfunction	1.30 (0.88–5.38)	0.38
Post–procedure MR	1.25 (0.89–2.34)	0.25
FMR vs. DMR	1.67 (0.98–3.74)	0.07

The values are mean ± SD or n (%). Abbreviations: STS, society of thoracic surgery; GFR, glomerular filtration rate; LVEF, left ventricular ejection fraction; NYHA, New York heart association; MR, mitral regurgitation; FMR, functional mitral regurgitation; DMR, degenerative mitral regurgitation.

**Table 5 jcm-11-06010-t005:** Cox regression for MACE.

Variable	HR (95% CI)	*p* Value
Age (years)	1.10 (1.01–1.24)	0.02
STS score	1.23 (0.97–1.54)	0.09
Diabetes Mellitus	1.39 (0.73–4.21)	0.28
GFR (mL/min)	0.97 (0.94–1.01)	0.07
LVEF (per 1%)	0.92 (0.84–0.98)	0.04
Functional Class (NYHA)	1.08 (0.84–2.93)	0.42
Right ventricular dysfunction	1.32 (0.82–5.12)	0.39
Post–procedure MR	1.96 (1.09–2.77)	0.03
FMR vs. DMR	1.78 (1.23–2.48)	0.04

The values are mean ± SD or n (%). Abbreviations: MACE, major adverse cardiovascular events; STS, society of thoracic surgery; GFR, glomerular filtration rate; LVEF, left ventricular ejection fraction; NYHA, New York heart association; MR, mitral regurgitation; FMR, functional mitral regurgitation; DMR, degenerative mitral regurgitation.

## Data Availability

The data underlying this article will be shared on reasonable request to the corresponding author.

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
