# Peer review of "Outcomes of Transcatheter Edge-to-Edge Repair in Degenerative vs. Functional Mitral Regurgitation"

_jcm, 2022, doi:10.3390/jcm11206010_

Round 1

Reviewer 1 Report (Previous Reviewer 2)

I appreciate the authors' efforts in revising their manuscript, according to the reviewers' comments.

Reviewer 2 Report (Previous Reviewer 1)

Comments and concerns have been adequately addressed.

This manuscript is a resubmission of an earlier submission. The following is a list of the peer review reports and author responses from that submission.

Round 1

Reviewer 1 Report

The authors are to be congratulated for a well-written manuscript.  I have a few discussion points for their consideration:

1.  The acronym MACE is not defined in the abstract - please put the acronym within line 29 where MACE is described.

2. While it is apparent from the results and Figure 1 that 17 patients were excluded due to meeting the exclusion criteria, including the n-number in the Methods section for those categories is confusing (since n=0 for the remaining categories).  Perhaps leave out the n-number in the Methods section as it is clear who was excluded and for what reason, and the n-numbers are given later in the Results section/cohort diagram.

3. Why were patients who underwent a previous MitraClip not excluded?  Is there any information on why the first MitraClip procedure failed and would that have any bearing on whether the second would be expected to succeed?

4. Is there a reason for giving TR grade and RV dysfunction grades as both descriptive and numerical? What criteria were used (please cite).

5. How was MR graded?  Please cite.  Were only patients with severe MR included or did you also include patients with moderate MR?  For follow-up, how was MR graded and what was the incidence of moderate and severe recurrent MR at follow-up?  Why wasn’t recurrence of >moderate MR considered an endpoint (either primary or secondary), or was that understood to be a part of MV surgery or TEER re-intervention?  If so, that should be stated (what was the criteria for MV surgery within 12m or TEER re-intervention within 12).  How many patients had recurrent moderate or severe MR, and of those patients, how many had a re-intervention? 

5. Can you give a breakdown of the type and n-number of DMR in the cohort (e.g. Carpentier class), and discuss whether or not in retrospect TEER was appropriate for those patients?

Author Response

The authors are to be congratulated for a well-written manuscript.  I have a few discussion points for their consideration:

Thank you for both the kind words and important notes.

  1. The acronym MACE is not defined in the abstract - please put the acronym within line 29 where MACE is described.

We've added the acronym "MACE".

  1. While it is apparent from the results and Figure 1 that 17 patients were excluded due to meeting the exclusion criteria, including the n-number in the Methods section for those categories is confusing (since n=0 for the remaining categories).  Perhaps leave out the n-number in the Methods section as it is clear who was excluded and for what reason, and the n-numbers are given later in the Results section/cohort diagram.

Thank you for this important note. We deleted both n-numbers form the methods section.

  1. Why were patients who underwent a previous MitraClip not excluded?  Is there any information on why the first MitraClip procedure failed and would that have any bearing on whether the second would be expected to succeed?

Thank you for these important comments/questions. In fact, all patients who entered this study were treated with MitraClip for the first time. A re-MitraClip was part of the MACE. This should have been pointed out. We have now added this to the first sentence in the methods (in the “Patients and data collection” paragraph): “In this prospective registry, all patients undergoing TEER for the first time from January 2012…”. As for failures, there were five patients who required re-intervention. This now was added to the end of the “Clinical Outcomes” paragraph in the results: “There were five patients who required surgery or re-intervention with a MitraClip during the follow-up period. One case in the DMR group was due to early dehiscence in a patient with Barlow’s disease and required surgery. Four patients in the FMR group required re-clip intervention during the 12-month follow up, due to further remodeling and dilatation of the left ventricle and the annulus.

4. Is there a reason for giving TR grade and RV dysfunction grades as both descriptive and numerical? What criteria were used (please cite).

Thank you for pointing this out. There really isn’t a reason for giving grades as both descriptive and numerical. We now removed the numbers from table 2 and left it descriptive only. As for the grading of TR, we have begun collecting data before the 5-point scale proposed by many, led by R. Hahn et al (1), which includes massive and torrential sub-categories within the previous definition of severe TR. We have based our initial estimations according to the older classification according to both the European Association of Cardiovascular Imaging (EACVI) and American Society of Echocardiography (ASE) recommendations, where “Severe TR” is defined quantitatively as an effective regurgitant orifice area (EROA) of ≥40 mm2 and a regurgitant volume of ≥45 ml (2,3). This point will be concisely added to the limitations.

Reference-

  1. Rebecca T Hahn, Jose L Zamorano, The need for a new tricuspid regurgitation grading scheme, European Heart Journal- Cardiovascular Imaging, Volume 18, Issue 12, December 2017, Pages 1342–1343.
  2. Patrizio Lancellotti, Christophe Tribouilloy, Andreas Hagendorff, Bogdan A. Popescu, Thor Edvardsen, Luc A. Pierard, Luigi Badano, Jose L. Zamorano, On behalf of the Scientific Document Committee of the European Association of Cardiovascular Imaging: Thor Edvardsen, Oliver Bruder, Bernard Cosyns, Erwan Donal, Raluca Dulgheru, Maurizio Galderisi, Patrizio Lancellotti, Denisa Muraru, Koen Nieman, Rosa Sicari, Document reviewers: Erwan Donal, Kristina Haugaa, Giovanni La Canna, Julien Magne, Edyta Plonska, Recommendations for the echocardiographic assessment of native valvular regurgitation: an executive summary from the European Association of Cardiovascular Imaging, European Heart Journal - Cardiovascular Imaging, Volume 14, Issue 7, July 2013, Pages 611–644.
  3. Zoghbi WA, Adams D, Bonow RO, et al. Recommendations for Noninvasive Evaluation of Native Valvular Regurgitation: A Report from the American Society of Echocardiography Developed in Collaboration with the Society for Cardiovascular Magnetic Resonance. J Am Soc Echocardiogr. 2017;30(4) Pages 303-371.

5a.
How was MR graded?  Please cite.  Were only patients with severe MR included or did you also include patients with moderate MR?  For follow-up, how was MR graded and what was the incidence of moderate and severe recurrent MR at follow-up?

MR was graded according to the above guidelines- Mild to severe, and only patients who were considered symptomatic severe MR, as per the guidelines, were included in this study. This description was added to the manuscript, so now the first sentence of the methods is: “In this prospective registry, included were all patients undergoing TEER at our institution for the first time using the MitraClip percutaneous mitral valve repair (Abbott Vascular, Inc., Santa Clara, California) at our institution, between January 2012 and May 2021. All patients suffered from severe symptomatic MR.”

5b. Why wasn’t recurrence of >moderate MR considered an endpoint (either primary or secondary), or was that understood to be a part of MV surgery or TEER re-intervention?  If so, that should be stated (what was the criteria for MV surgery within 12m or TEER re-intervention within 12m).

Thank you for this important remark. We have now added this sentence to the paragraph related to the endpoints: “The co-primary endpoints were rates of all-cause death and major adverse cardiac events (MACE, which comprised: all-cause death, hospitalizations for heart failure, mitral valve surgery or TEER re-interventions) at 12-month follow-up. Patients were treated by mitral valve surgery or TEER re-intervention during follow-up if recurrent symptomatic moderate MR or above, amenable to the appropriate therapy, was demonstrated. Secondary outcomes included the individual components of MACE and rates of moderate MR or above”

5c. How many patients had recurrent moderate or severe MR, and of those patients, how many had a re-intervention? 

As mentioned above (in the response to question 3), there were five patients who required surgery or re-intervention with a MitraClip during the follow-up period. In addition, there were three patients in the FMR group and one in the DMR group who had estimated moderate MR but were relatively asymptomatic (NYHA 1-2) and were treated medically. This too was now added to the relevant paragraph in the “Clinical Outcomes” paragraph.

  1. Can you give a breakdown of the type and n-number of DMR in the cohort (e.g. Carpentier class), and discuss whether or not in retrospect TEER was appropriate for those patients?

We have re-assessed the exact type of both DMR and FMR mechanisms as best as possible. We have identified patients according to the Carpentier classification, as well as by the division to “ventricular functional” (primarily) or “atrial functional” MR. As expected, all 31 patients in the DMR group belonged to the Carpentier type II, where a flail mitral valve was seen in 22, and 9 had mitral valve prolapse. Of these, 4 were considered true Barlow’s disease, the others due to fibroelastic deficiency. We had no Carpentier type I or IIIa cases in the DMR group. As for the FMR group, we estimate that 68 of the patients fit the Carpentier type IIIb, whereas 9 had mostly atrial/annular dilatation in the context of atrial fibrillation and heart failure with preserved ejection fraction, fitting the description of atrial functional MR and Carpentier type I (ref 1, 2). An analysis for differences in the effect of TEER on outcomes according to the different Carpentier subtypes was found to be non-significant (p=0.274 in the DMR group and p=0.482 in the FMR group).

Reference-

  1. Deferm S, Bertrand P, Verbrugge F, et al. Atrial Functional Mitral Regurgitation. J Am Coll Cardiol. 2019 May, 73 (19) 2465–2476.
  2. El Sabbagh A, Reddy YNV, Nishimura RA. Mitral Valve Regurgitation in the Contemporary Era: Insights Into Diagnosis, Management, and Future Directions. JACC Cardiovasc Imaging. 2018;11(4):628-643.

Reviewer 2 Report

Using their institutional database between January 2012 and May 2021 involving 108 patients undergoing transcatheter edge-to-edge repair (TEER) through the application of MitraClip technology, Kheifets and colleagues were able to show that higher rates of both death and MACE were observed in the functional mitral regurgitation (FMR, n=77) group when compared with the degenerative mitral regurgitation (DMR, n=31) group. Such findings were indeed interesting and timely.

My major concern is related to the inadequate statistical power owing to the small sample sizes (particularly that of the DMR group), which made the current comparison and conclusion less convincing.

It appeared unclear regarding the authors’ definition of procedural “success” in the two groups of patients studied. Moreover, with the lacking of both early postoperative and 1-year echo data, the current study provided little (if any) new evidence in guiding our future clinical practice.

Some other clinical details may deserve further clarification by the authors. For instance, the possible impact of atrial FMR vs ventricular FMR (i.e., ischemic MR or MR secondary to dilated cardiomyopathy) were unclear in the current study. Since greater incidence of atrial fibrillation and higher pulmonary artery pressure were observed in the FMR group, whereas no inter-group differences in terms of the LVEDd measurements, the potential influence of atrial FMR on survival and MACE should also be analysed.

Author Response

Using their institutional database between January 2012 and May 2021 involving 108 patients undergoing transcatheter edge-to-edge repair (TEER) through the application of MitraClip technologyKheifets and colleagues were able to show that higher rates of both death and MACE were observed in the functional mitral regurgitation (FMR, n=77) group when compared with the degenerative mitral regurgitation (DMR, n=31) group. Such findings were indeed interesting and timely.

Thank you for the kind words.

  1. My major concern is related to theinadequate statistical power owing to the small sample sizes (particularly that of the DMR group), which made the current comparison and conclusion less convincing.

Thank you for this important point. The small sample size, particularly that of the DMR group, is in fact a limitation. Hence, it was highlighted it in the limitation section of out paper.

  1. It appeared unclear regarding the authors’ definition of procedural “success” in the two groups of patients studied. Moreover, with the lacking of both early postoperative and 1-year echo data, the current study provided little (if any) new evidence in guiding our future clinical practice.

Thank you for this important comment. Indeed, we can and should elaborate: our definition of “successful” TEER procedure is mild or less residual MR (≤1+), as was recently validated and adopted in several medical centers worldwide (ref 1-3). This was now added to the methods section.

 Reference-

  1. Sugiura A, Kavsur R, Spieker M, et al. Recurrent Mitral Regurgitation After MitraClip: Predictive Factors, Morphology, and Clinical Implication. Circ Cardiovasc Interv. 2022;15(3):e010895.
  2. Buzzatti N, De Bonis M, Denti P, et al. What is a "good" result after transcatheter mitral repair? Impact of 2+ residual mitral regurgitation. J Thorac Cardiovasc Surg. 2016;151(1):88-96.
  3. Hassan A, Eleid MF. Recurrent Mitral Regurgitation After MitraClip: Defining Success and Predicting Outcomes. Circ Cardiovasc Interv. 2022;15(3):e011837.

With regards to the lack of data of postoperative echocardiography, we have now also added information on the post-operative MR, both at the immediate phase, after 1 month and at 12 months (please see the revised table 3).

  1. Some other clinical details may deserve further clarification by the authors. For instance, the possible impact of atrial FMRvs ventricular FMR (i.e., ischemic MR or MR secondary to dilated cardiomyopathy) were unclear in the current study. Since greater incidence of atrial fibrillation and higher pulmonary artery pressure were observed in the FMR group, whereas no inter-group differences in terms of the LVEDd measurements, the potential influence of atrial FMR on survival and MACE should also be analysed.

Thank you for raising this important point. As mentioned in the response to reviewer 1, we have re-assessed the exact type of both DMR and FMR mechanisms as best as possible. We have identified patients according to the Carpentier classification, as well as by the division to “ventricular functional” (primarily) or “atrial functional” MR. As expected, all 31 patients in the DMR group belonged to the Carpentier type II, where a flail mitral valve was seen in 22, and 9 had mitral valve prolapse. Of these, 4 were considered true Barlow’s disease, the others due to fibroelastic deficiency. We had no Carpentier type I or IIIa cases in the DMR group. As for the FMR group, we estimate that 68 of the patients fit the Carpentier type IIIb, whereas 9 had mostly atrial/annular dilatation in the context of atrial fibrillation and heart failure with preserved ejection fraction, fitting the description of atrial functional MR and Carpentier type I (ref 1, 2). An analysis for differences in the effect of TEER on outcomes according to the different Carpentier subtypes was found to be non-significant (p=0.274 in the DMR group and p=0.482 in the FMR group).

Reference-

  1. Deferm S, Bertrand P, Verbrugge F, et al. Atrial Functional Mitral Regurgitation. J Am Coll Cardiol. 2019 May, 73 (19) 2465–2476.
  2. El Sabbagh A, Reddy YNV, Nishimura RA. Mitral Valve Regurgitation in the Contemporary Era: Insights Into Diagnosis, Management, and Future Directions. JACC Cardiovasc Imaging. 2018;11(4):628-643.

Round 2

Reviewer 2 Report

The authors' efforts in revising their manuscript are recognised by this reviewer. The lack of adequate statistical power is indeed an issue that should not be ignored. The current conclusion was unfortunately not well supported by the data as presented.